# Performance Study on the Effect of Coolant Inlet Conditions for a 20 Ah LiFePO$_4$ Prismatic Battery with Commercial Mini Channel Cold Plates

Jeevan Jaidi [1,*], Sandeep Dattu Chitta [1], Chaithanya Akkaldevi [1], Satyam Panchal [2], Michael Fowler [3] and Roydon Fraser [2]

1   Department of Mechanical Engineering, BITS Pilani, Hyderabad Campus, Secunderbad 500078, India; shanmukhasandeep98@gmail.com (S.D.C.); chaithanya.akkaldevi3@gmail.com (C.A.)
2   Mechanical and Mechatronics Engineering Department, University of Waterloo, Waterloo, ON N2L 3G1, Canada; satyam.panchal@uwaterloo.ca (S.P.); rafraser@uwaterloo.ca (R.F.)
3   Chemical Engineering Department, University of Waterloo, Waterloo, ON N2L 3G1, Canada; mfowler@uwaterloo.ca
*   Correspondence: jaidi@hyderabad.bits-pilani.ac.in; Tel.: +91-40-66303548

**Abstract:** Rechargeable Li-ion batteries are widely used in renewable energy storage and automotive powertrain systems, and therefore, an efficient thermal management system is imperative for maximum battery life and safety. Battery heat generation and dissipation rates primarily depend on the battery surface temperatures, which are affected by the coolant system design and coolant inlet conditions. In this paper, a two-way coupled electrochemical-thermal simulation with selected experimental validation has been performed and analyzed the effect of water coolant inlet conditions on the effectiveness of commercial mini-channel cold-plates for 20 Ah LiFePO$_4$ prismatic batteries. Three coolant inlet temperatures (25–45 °C) and four flow rates (150–600 mL/min) are tested at three different discharge rates (2–4 C) and the performance of coolant system design has been analyzed in terms of battery peak (maximum) temperature and temperature difference (i.e., non-uniformity) across the battery. The predicted results indicate that the coolant flow rate has a profound effect on the battery temperature non-uniformity, while the coolant inlet temperature has a significant effect on the battery peak temperature. At high coolant flow rates, the battery surface temperature difference is within the acceptable range ($\Delta T < 5$ °C), but the maximum temperatures are high at all discharge rates. Further, at the low coolant inlet temperature of 25 °C and the high coolant flow rate of 600 mL/min, the battery temperature rise at the top and bottom locations during the constant current discharge process is high, indicating that the battery heat generation rate is high at a low coolant inlet temperature.

**Keywords:** Li-ion battery; mini-channel cold-plates; coolant flow rate; peak temperature; temperature non-uniformity; electrochemical-thermal model; COMSOL software

## 1. Introduction and Literature Review

Though the development and deployment of Li-ion battery (LIB) packs for renewable energy storage and electric vehicles (EVs) have taken place across the globe, there are serious concerns, such as thermal runaway, capacity and cycle life degradation, etc. Researchers have shown that the performance of LIBs strongly depends on battery operating temperatures (represented in terms of two critical temperatures—maximum temperature, $T_{max}$, and temperature difference across the battery surface, $\Delta T$) and suggested an optimum range ($T_{max}$: 15–35 °C; $\Delta T < 5$ °C) for the maximum battery capacity and minimal degradation [1]. These critical temperatures are influenced by the battery discharge rates and coolant flow conditions (flow rate and inlet temperature). For instance, an experimental test showed a 22.5% increase in discharge time at 25 °C operating temperature in contrast to 50 °C [2].

Further, a battery pack of three 10 Ah LiFePO4 cells sandwiched between the cold-plates are tested at different discharge rates and coolant inlet temperatures. The authors found that at 10 °C the coolant inlet and at all C-rates, the battery surface maximum temperature was below the optimum range (25–40 °C) and concluded that that the low temperatures lowered the battery capacity and life [3]. Therefore, to design an efficient Battery Thermal Management System (BTMS), an in-depth understanding of battery heat generation and dissipation rates at a battery level is very essential. The following paragraphs discuss the recent work conducted by various researchers on the measured and predicted heat generation rates, coolant system designs and coolant inlet conditions (temperature and flow rate) used at different charge and discharge rates.

The measured heat generation rates of 20 Ah $LiFePO_4$ prismatic battery have shown that heat generation increases with increase in discharge rates and decrease in operating temperatures [4]. The measured heat generation rates of 8 Ah $LiMn_2O_4$ battery during charge–discharge cycles have shown that joule heating increases with decrease in battery temperatures, while heat of reaction does not vary significantly [5]. The measured and predicted voltage and temperature of $LiFePO_4$ battery have shown that reaction heat majorly contributes to the battery heat generation, followed by contact resistance heat and joule heat [6]. The heat generation rates of 2.6 Ah $LiFePO_4$ cylindrical battery were predicted at different rates based on the measured battery temperatures and heat fluxes by making an energy balance [7]. A review on uncoupled and coupled battery-thermal models stated that the lack of measured data of several parameters in real-time conditions as well as assumptions of constant and uniform properties made the P2D model qualitative. Henceforth, predicted temperatures cannot match with the measured data [8]. Considering, overpotential and entropic heats with negligible heat of mixing, the predicted heat generation rates of 45 Ah $LiFePO_4$ pouch batteries are higher during discharge than charge. Therefore, the authors suggested that an effective cooling system is a must to prevent thermal runaway at high C-rates [9]. A coupled P2D-thermal model developed in COMSOL software for 4 Ah Li-ion NCA/graphite battery predicted that the maximum temperature is near the tabs and positive current collector and is attributed to high ohmic heat generation [10]. The predicted heat generation rates of a 19.5 Ah $LiFePO_4$ prismatic battery using overpotential and voltage difference methods have indicated similar results at all state-of-charge (SOC) and discharge rates with the difference being less than 3% [11]. Further, different cooling methods compared has shown that water cooling is the best, followed by water plus foam, PCM, foam and air. A review on the effect of ambient temperatures and cooling methods for battery thermal management stated that water-cooling has been widely researched, and cold-plates are used where space is a constraint [12]. The predicted voltage and thermal responses of 20 Ah $LiFePO_4$ prismatic batteries using single- and multi-layer approaches were compared with measurements at an ambient temperature (25 °C) and different discharge rates [13]. The authors stated that a cooling system is indispensable to maintain battery temperatures within the permissible range. A study on the effectiveness of the air and liquid cooling of Li-ion battery packs has concluded that water is three times more effective in cooling than air for similar rise in battery temperature [14].

A 20 Ah $LiFePO_4$ prismatic battery placed between cold-plates having single mini-channel with multiple turns was experimentally studied at different discharge rates and coolant inlet temperatures [15,16]. Additionally, ten thermocouples were fixed across the battery and measured the temperature rise during the discharge. The authors found that battery peak (maximum) temperature increased with increasing C-rate and coolant inlet temperature. A drive-cycle test with a pack of three 10 Ah $LiFePO_4$ batteries (in series and sandwiched between cold-plates) was conducted at different discharge rates and coolant inlet temperatures, and found that battery peak temperatures within the permissible range [3]. A conjugate heat transfer model used for pouch type Li-ion battery pack with coolant channels between the batteries has concluded that the coolant flow rate significantly affects the battery peak temperature at higher C-rates than at lower C-rates [17]. A numerical study on the effectiveness of the liquid cooling of a 55 Ah lithium-ion battery,

with different channel configurations, flow rates, and flow direction, has shown increased thermal performance with an increase in the number of channels and inlets and outlets on the same side of the battery [18]. A numerical study on a battery module of 7 Ah lithium-ion batteries sandwiched between aluminium cold-plates and discharged at 5 C has shown that the coolant flow direction towards the electrode showed the best performance, and coolant flow rate has a limited effect on the battery temperatures [19]. The coupled P2D-thermal model developed for 20 Ah LiFePO$_4$ prismatic battery, using COMSOL Multiphysics software, has shown that the multi-layer-based model results are in very close agreement with the measured data than the single-layer model [20]. A coupled P2D-thermal model for 10 Ah LiFePO$_4$ battery was developed by Lai et al. [21], and accounted the reaction heat, the ohmic heat and the active polarization heat. The reversible and irreversible heat generations were predicted at different discharge rates and found that the latter is relatively more stable than the former. Panchal et al. [22] did experiments and performed simulations (using ANSYS software) on 20 Ah LiFePO$_4$ prismatic batteries sandwiched between cold-plates having a single mini-channel with five U-turns and inlet and outlet on same side of the battery. Battery surface temperatures measured at different coolant inlet temperatures and discharge rates. The authors observed a loss in battery discharge capacity at low coolant inlet temperatures. The performance of 20 Ah LiFePO$_4$ prismatic battery with silica-liquid cooling plates (SLCPs) was studied by Wang et al. [23], and varied the number of channels, flow rates and flow directions. However, measured heat generation rates were used in the simulations. The authors concluded that increasing the coolant flow rate beyond a certain limit has no significant effect on battery temperatures. Yuan et al. [24] numerically studied the liquid cooling and heating of a battery module (three lines and four rows) with a cooling jacket placed between the lines. However, a heat flux value was used in the simulations.

It can be noted from the above literature survey that the battery operating temperatures are sensitive to the discharge rates, coolant system design (single vs. multi-channels), coolant flow direction (inlet and outlet positions) and coolant inlet conditions (temperature and flow rate). However, there is no single coolant system tested with the best design options for wide range of coolant inlet conditions and discharge rates. Recently, the present authors have numerically and experimentally studied on the thermal performance of commercial mini-channel cold-plates for 20 Ah LiFePO$_4$ prismatic battery at different discharge rates and coolant inlet temperatures for a fixed coolant flow rate of 150 mL/min [25,26]. In the present work, the effect of coolant flow rates (150, 300, 450 and 600 mL/min) have been numerically studied and analyzed at different combinations of coolant inlet temperatures and discharge rates to which the heat generation rate and battery temperatures are sensitive. It can be noted that increasing the coolant flow rate from 150 mL/min to 300 mL/min, and above, results in enhanced heat dissipation from the battery because of turbulent convection in the mini-channels, and hence one would expect significant effect on the battery surface temperatures, temperature uniformity and heating rate.

## 2. Modelling and Simulations

In this section, the geometrical configuration details of 20 Ah LiFePO$_4$ prismatic batteries and commercial mini-channel cold-plates, together as a Battery Thermal Management System (BTMS), are discussed along with meshing created for numerical simulations. Additionally, discussed briefly on different phenomena involved in battery and thermal models along with the governing equations and numerical solvers used in COMSOL software. Further, the coupling of battery and thermal models and the selected experiments performed for validation of model results are briefly discussed and referred to present authors' recent papers for more details.

### 2.1. BTMS—Geometry, Mesh and Experiments

Battery Thermal Management System (BTMS) considered for the present study consists of a 20 Ah LiFePO$_4$ prismatic battery placed between two commercial cold-plates with

coolant flow distributed through mini-channels having a single U-turn and inlet and outlet across the battery, as shown in Figure 1. The dimensions of principal surface of battery and cold-plates are 157 mm × 227 mm, while the thickness of battery and cold-plates are 7.25 mm and 1.7 mm, respectively. The cross-section of individual mini-channels is hexagonal with a width equal to 2 mm. Due to plane symmetry of the battery and cold-plate arrangement, and also to reduce the computational time, one cold-plate with half a battery in thickness direction is used for the numerical simulations with symmetry boundary condition on the respective principal surface. In numerical simulations, meshing plays a vital role and depending upon the physics and gradients of variables involved, a mesh independent exercise must be done. Accordingly, the element sizes in solid and fluid domains are arrived. A tetrahedral mesh for the battery and cold-plate and hexagonal mesh for the tabs are used. For fluid domain in mini-channels, the minimum and maximum mesh sizes used are 0.25 mm and 0.625 mm, respectively. Further, to resolve the viscous effects of flow at the channel surfaces, four layers are used. A total mesh (element) count of 5.4 million is used in all the simulations discussed in this paper. For more details on the mesh independent study conducted by testing on the battery surface peak temperature, it is suggested to refer to the recent papers by present authors [25,26]. For the purpose of quantitative validation of the present coupled model predictions, the available measured data on voltage-capacity variation during discharge and thermal response by the Thermal Management System (TMS) at a fixed coolant flow rate of 150 mL/min and different coolant inlet temperatures and discharge rates have been used. For more details on the test bench developed at the University of Waterloo, Canada, and the test procedure followed during the measurements are discussed in the recent paper by present authors [25].

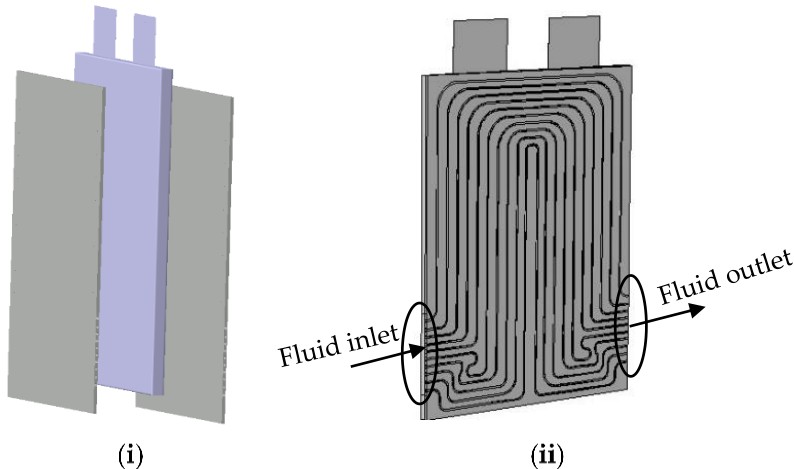

**(i)** **(ii)**

**Figure 1.** (**i**) Battery with cold-plates, and (**ii**) mini-channels inside a cold-plate.

## 2.2. BTMS—Phenomena and Governing Equations

To design a Battery Thermal Management System (BTMS), a detailed knowledge on the sources of heat generation in a battery as well as the heat dissipation mechanisms by the coolant system design are required. Three sources of heat generation in Li-ion batteries, namely overpotential heat, entropy change heat and heat of mixing have been identified and used in the analysis of BTMS. However, the heat of mixing is found to be less significant among the heat sources, and therefore the total heat generation considered in the battery (P2D) model is attributed to irreversible overpotential (ohmic and polarization) and reversible entropic change heats. In the present simulations' study, a 1D electrochemical (P2D)-thermal model coupled with a 3D conjugate heat transfer model developed in COM-SOL Multiphysics software is used to study the effectiveness of a commercial mini-channel cold-plate design for prismatic batteries at different coolant inlet conditions (laminar and turbulent flows) at different discharge rates typically used in EVs. It can be noted that the battery (P2D)–thermal coupling is performed to update the instantaneous average

heat generation and battery temperature taking into account the temperature dependent battery model parameters. For more details on the coupling of battery–thermal and the conjugate heat transfer models along with assumptions invoked, it is suggested to refer to recent papers by the present authors [25,26]. The following sub-subsections discuss on the governing equations used in the above models.

### 2.2.1. Li-Ion Battery Model

Li-ion battery (electrochemical) model is the accurate one among different theoretical models available in the literature. The electrochemical model (P2D) considers many theories, including porous electrode, Ohm's law, mass transfer in solid and electrolyte phases, concentrated solution, and intercalation/deintercalation kinetics. Thus, it solves a number of PDEs and requires large input data, which are specific to the type of battery and battery temperatures. The electrochemical process input parameters required includes the diffusion coefficient of $Li^+$ in solid and electrolyte phases, the ionic electrical conductivity in the electrolyte phase, the entropy coefficient, and the reaction rate. The electrochemical model provides four variables, namely solid-phase potential ($\varphi_s$), electrolyte-phase potential ($\varphi_e$), concentration of $Li^+$ in solid-phase ($c_s$) and concentration of $Li^+$ in electrolyte-phase ($c_e$) by solving the following conservation equations of charge and species in the solid- and electrolyte phases. Butler–Volmer kinetics is used to determine reaction rate and also to couple the conservation equations [21]. For detailed nomenclature of various parameters, variables, constants and boundary conditions used in P2D model can be referred in the literature [27–29].

- Mass conservation of $Li^+$ species in a solid-phase

$$\tau^2 R_i \frac{c_{s,i}}{\partial t} = \frac{D_{s,i}}{R_i}\left[\frac{\partial}{\partial \tau}\left(\tau^2 \frac{ac_{s,i}}{\partial \tau}\right)\right] \tag{1}$$

- Mass conservation of $Li^+$ species in electrolyte-phase

$$\varepsilon_e \frac{dc_e}{dt} + \nabla \cdot \left\{-D_e^{eff}\nabla c_e\right\} = \frac{S_{a,i}j_{loc,i}}{F}(1 - t_+) \tag{2}$$

where, $D_e^{eff} = D_e \varepsilon_e^{\gamma_e}$

- Electronic charge transport in a solid-phase:

$$\nabla \cdot (-\sigma_c \nabla \varphi_c) = -J_i \tag{3}$$

$$\nabla \cdot \left(-k_s^{eff}\nabla \varphi_{c_s}\right) = -S_{a,i}\left(j_{loc,i} + C_{dl}\left(\frac{\partial \varphi_{c_s}}{\partial t} - \frac{\partial \varphi_{c_e}}{\partial t}\right)\right)$$
$$S_{a,i} = \frac{3\varepsilon_{s,i}}{r_{p,s}}; k_s^{eff} = k_s \varepsilon_s^{\gamma_s} \tag{4}$$

- Electronic charge transport in electrolyte-phase:

$$\nabla \cdot \left\{k_e^{eff}\left[-\nabla \varphi_{c_e} + \frac{2RT}{F}\left[1 + \frac{\partial \ln f}{\partial \ln c_e}\right](1 - t_+)\frac{\nabla c_e}{c_e}\right]\right\} = S_{a,j}j_{loc,i} \tag{5}$$

- Electrochemical kinetics:

$$j_{loc,i} = j_{0,i}\left\{exp\left[\frac{\alpha_{a,i}\eta_i F}{RT}\right] - exp\left[\frac{-\alpha_{ci}v_i F}{RT}\right]\right\} \tag{6}$$

where,

$$\eta_i = \varphi_{c_{s,i}} - \varphi_{c_{e,i}} - U_i$$
$$j_{0,i} = Fk_i c_e^{\alpha_{ai}}(c_{s,max,i} - c_{s, surf ,i})^{\alpha_{ai}} c_{s, surf ,i}^{\alpha_{c,i}}$$

2.2.2. Conjugate Heat Transfer Model

As part of the Thermal Management System (TMS), a 3D conjugate heat transfer model is developed and used for heat dissipation from the battery—firstly, by diffusion from the battery primary surface into the solid part of the cold-plates, and secondly, by convection through coolant flow in the mini-channels embedded in the cold-plates. Since the objective of present study is to test the efficacy of coolant system design by different coolant inlet conditions (temperatures and flow rates), four different flow rates (150, 300, 450 and 600 mL/min) are considered. At a low coolant flow rate of 150 mL/min, the flow is laminar and at all other flow rates (300–600 mL/min) the flow is turbulent. Depending on the coolant flow rate considered, either laminar or turbulent flow equations are solved along with the energy equation, as part of a 3D conjugate heat transfer model for a BTMS coolant system design.

A low Reynolds number based two-equation turbulence model (referred to as modified $k - \varepsilon$ model) is used in the present simulations. The modifications are done through the wall function coefficients. Several such modified turbulence models are available in the literature and each model is specific to different situations wherein a flow separation takes place in duct bends with small or large curvature. In the present study, an AKN model (named after the pioneers: Abe, Kondoh and Nagano) available in the COMSOL Multiphysics software is used. Overall, the 3D conjugate heat transfer model solves the following transient incompressible fluid flow and energy equations.

- Conservation of mass:

$$\rho \nabla \cdot \boldsymbol{u} = 0 \tag{7}$$

- Conservation of momentum (laminar flow):

$$\rho \frac{\partial \boldsymbol{u}}{\partial t} + \rho (\boldsymbol{u} \cdot \nabla) \boldsymbol{u} = \nabla \cdot \left[ -p\boldsymbol{I} + \mu \left( \nabla \boldsymbol{u} + (\nabla \boldsymbol{u})^T \right) \right] \tag{8}$$

where $\boldsymbol{u}$ is velocity vector, $p$ is pressure, and $\mu$ and $\rho$ are the viscosity and density of coolant.

- Conservation of momentum (turbulent flow):

$$\rho \frac{\partial \boldsymbol{u}}{\partial t} + \rho \boldsymbol{u} \cdot \nabla \boldsymbol{u} + \nabla \cdot \overline{\left( \rho \boldsymbol{u}'(\boldsymbol{u}')^T \right)} = -\nabla P + \nabla \cdot \mu \left( \nabla \boldsymbol{u} + (\nabla \boldsymbol{u})^T \right) \tag{9}$$

The transport of turbulent kinetic energy and its dissipation rate, respectively, are expressed as

$$\rho \frac{\partial k}{\partial t} + \rho \boldsymbol{u} \cdot \nabla k = \nabla \cdot \left( \left( \mu + \frac{\mu_T}{\sigma_k} \right) \nabla k \right) + P_k - \rho \varepsilon \tag{10}$$

$$\rho \frac{\partial \varepsilon}{\partial t} + \rho \boldsymbol{u} \cdot \nabla \varepsilon = \nabla \cdot \left( \left( \mu + \frac{\mu_T}{\sigma_\varepsilon} \right) \nabla \varepsilon \right) + C_{\varepsilon 1} \frac{\varepsilon}{k} P_k - f_\varepsilon C_{\varepsilon 2} \rho \frac{\varepsilon^2}{k} \tag{11}$$

where $P_k$ is turbulent kinetic energy generation by shear and is expressed as

$$P_k = \mu_T \left( \nabla \boldsymbol{u} : \left( \nabla \boldsymbol{u} + (\nabla \boldsymbol{u})^T \right) - \frac{2}{3} (\nabla \cdot \boldsymbol{u})^2 \right) - \frac{2}{3} \rho k \nabla \cdot \boldsymbol{u} \tag{12}$$

and $\mu_T$ is the turbulent (eddy) viscosity, $R_t$ is the turbulent Reynolds number, and $f_\mu$ and $f_\varepsilon$ are modified wall functions, and are expressed as

$$\mu_T = \rho f_\mu C_\mu \frac{k^2}{\varepsilon} \tag{13}$$

$$R_t = \rho k^{\frac{2}{\mu \varepsilon}} u_\varepsilon = (\mu \varepsilon / \rho)^{\frac{1}{4}} \tag{14}$$

$$f_\mu = \left(1 - e^{-\frac{l'}{14}}\right)^2 \cdot \left(1 + \frac{5}{R_t^{\frac{3}{4}}} e^{-\left(\frac{R_t}{200}\right)^2}\right) \tag{15}$$

$$f_\varepsilon = \left(1 - e^{-\frac{l^*}{3.1}}\right)^2 \cdot \left(1 - 0.3 e^{-\left(\frac{R_t}{6.5}\right)^2}\right) \tag{16}$$

$$l^\star = (\rho u_\varepsilon l_w)/\mu \tag{17}$$

- Conservation of energy:

$$\frac{\partial (\rho C_p T)}{\partial t} + \boldsymbol{u}.\nabla (\rho C_p T) = \nabla.(k \nabla T) + \dot{q}_h''' \tag{18}$$

where $C_p$ is specific heat, $k$ is thermal conductivity, $\dot{q}_h''' \left(= \dot{Q}_h / \forall\right)$ is volumetric battery heat generation rate, and $\forall$ is battery volume.

The instantaneous average heat generation rate obtained from the battery (P2D) model is diffused into the solid cold-plates in thickness direction and then carried away by the coolant flow in mini-channels. It can be noted here that the above equations (Equations (7)–(18)) are generalized fluid flow and heat transfer equations, and depending on the domain (battery/cold-plates), domain type (solid/fluid) and transport phenomena involved (diffusion/laminar or turbulent convection) the COMSOL software selects and solves the simplified equations. For example, the energy equation for a battery as solid domain solves Equation (18) without convection (second term on the left-hand side). Similarly, the energy equation for cold-plates with mini-channels solves Equation (18) without a volumetric heat source (last term on the right-hand side). The heat generation rate ($\dot{Q}_h$) within the battery is calculated based on the energy conservation for a cell [30] and neglected the enthalpy of mixing and phase change. Further, depending on the flow type (laminar/turbulent) within the channels, continuity, momentum and energy equations are selected and solved.

### 2.3. Numerical Solvers in COMSOL Software

It should be noted that finite heat transfer occurs in battery and cold-plates (solid domains) as well as mini-channels (fluid domain). Fluid flow (laminar/turbulent) equations are solved using an algebraic multigrid solver and the heat transfer equation is solved using PARDISO solver. For time stepping, the Backward Differentiation Formula (BDF) method is used because of its stability and versatility.

### 3. Results and Discussion

A Battery Thermal Management System (BTMS), consisting of a 20 Ah LiFePO$_4$ prismatic battery with two mini-channel cold-plates (computational domain), mesh and the coupled Li-ion battery model (P2D) with input data used, is available in the authors' recently published paper [25]. Also, the battery electrical response in terms of variation of voltage-discharge capacity, predicted and measured voltage-time at different discharge rates are available, and hence, the same are not discussed here for brevity and simplicity. In this section, the effect of the water coolant inlet conditions (temperature and flow rate) on the predicted temperature distribution, peak temperature, and temperature difference across the battery surface at different discharge rates are discussed in detail. It can be noted that for the coolant channel design considered in the present study, the flow in individual channels is laminar at a coolant flow rate of 150 mL/min, while the flow is turbulent at higher flow rates (300, 450 and 600 mL/min). As expected with the turbulent flow, the mixing of coolant across the flow direction enhances the heat transfer in contrast to the laminar case, which in turn lowers the battery surface temperatures. However, due to increased flow rate, the coolant residence time between inlet and outlet decreases, which would result in no significant cooling of the battery surface beyond a certain flow rate but

the parasitic power requirement to supply the coolant increases. Henceforth, a trade-off between the coolant flow rate and battery surface temperatures must be considered in the coolant system design.

The effect of the water coolant flow rates (150–600 mL/min) on battery surface temperatures at 3 C and 4 C discharge rates with coolant inlet temperature at 25 °C are shown in Figures 2 and 3. It is clearly observed that at low coolant flow rate of 150 mL/min, the flow is laminar and the water at the core is less heated as compared to water at the channel surface. This resulted in a high peak temperature of about 33.02 °C and 36.88 °C for 3 C and 4 C discharge, respectively. With an increase in the coolant flow rate, the flow changes from laminar to turbulent, resulting in more uniform temperatures across the battery surface. For instance, the peak temperature dropped by about 3.48 °C and 4.63 °C for 3 C and 4 C discharge by increasing the flow rate from 150 to 300 mL/min. However, the drop in peak temperature is only about 0.65 °C and 1.01 °C for 3 C and 4 C discharge by increasing the coolant flow rate from 450 to 600 mL/min. This is because the viscous boundary layer thickness on the channel surface decreases with the increase in the coolant flow rate, and thereby less heat is absorbed by the coolant from channel surfaces. Further, the increase in the fluid velocity with an increased coolant flow rate results in less time for coolant to absorb the heat from channel surfaces. Therefore, from the above results and arguments, it can be stated that increasing the coolant flow rate beyond a certain limit does not result in a significant change in the battery surface peak temperature, but the parasitic power requirement to supply the coolant increases.

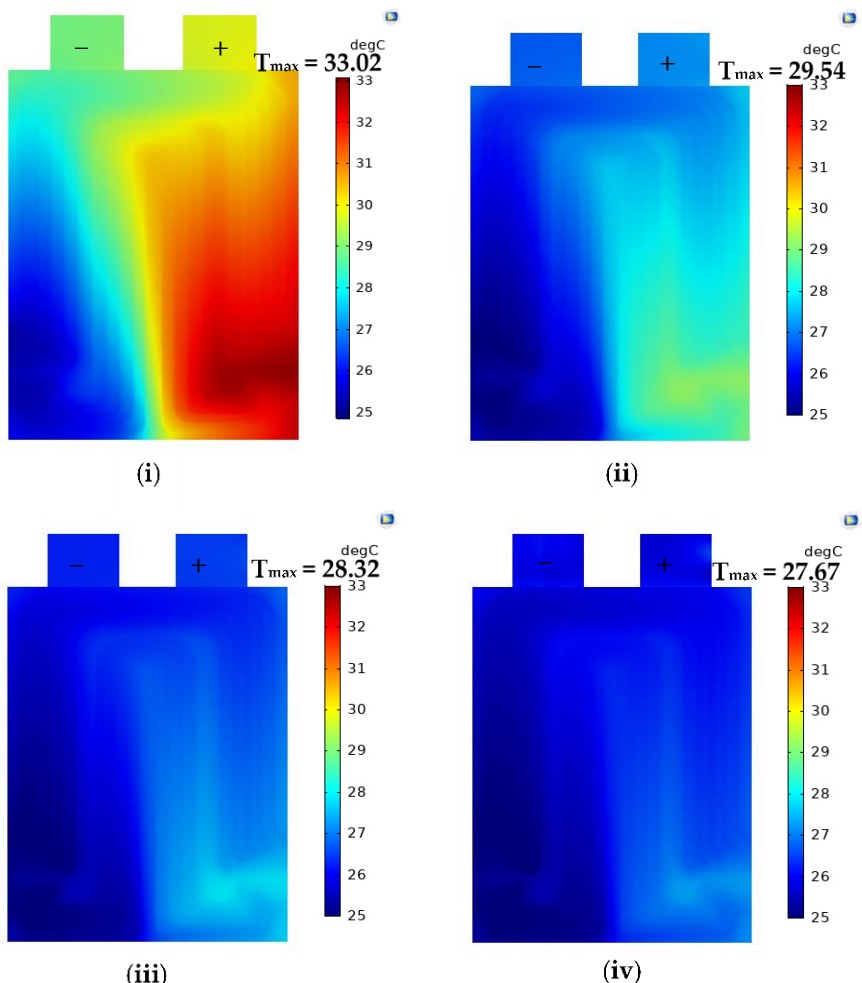

**Figure 2.** Predicted battery surface temperatures at 3 C discharge rate, 25 °C water inlet, and different flow rates, mL/min; (**i**) 150 (**ii**) 300 (**iii**) 450 (**iv**) 600.

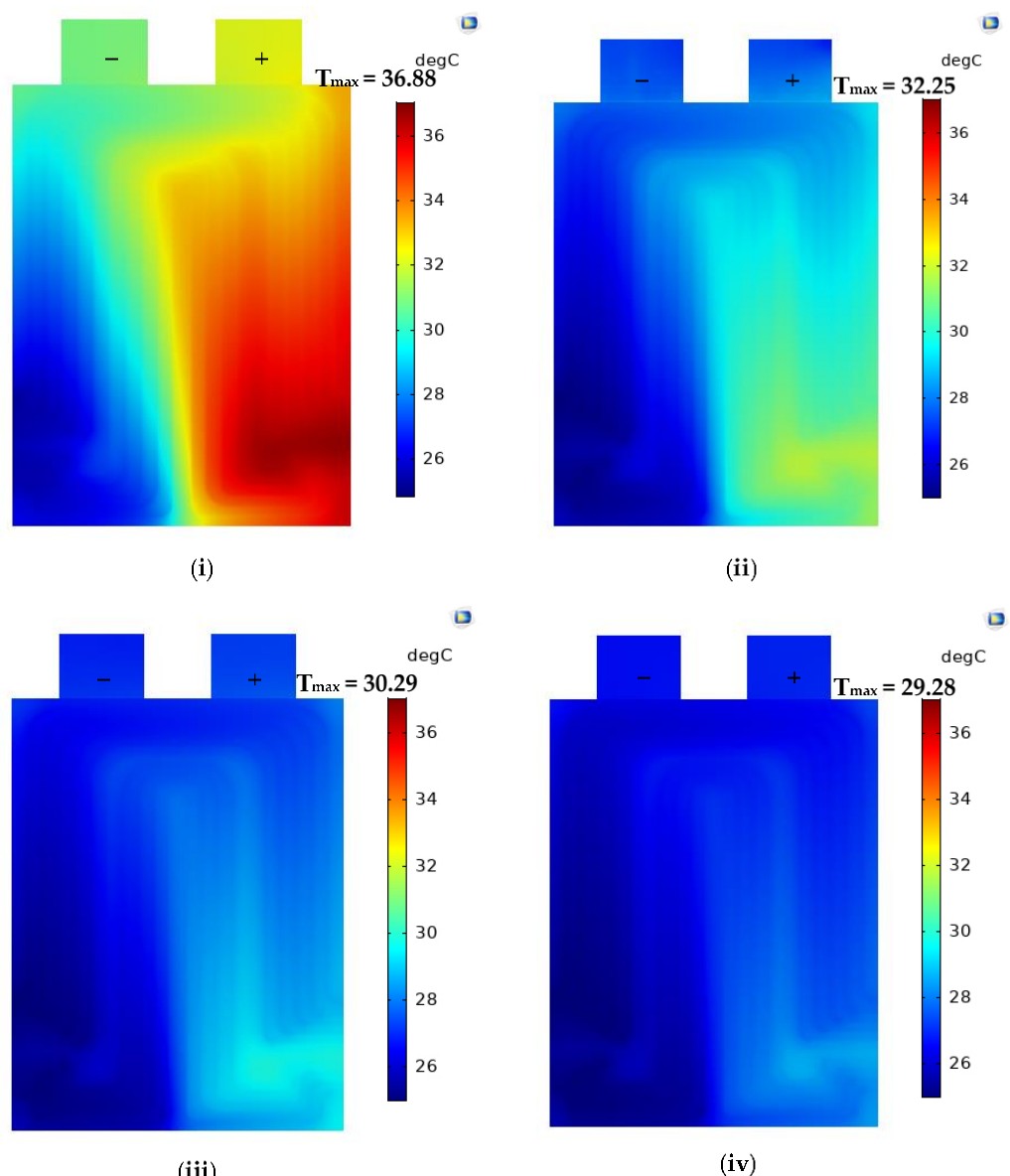

**Figure 3.** Predicted battery surface temperatures at 4 C discharge rate, 25 °C water inlet, and different flow rates, mL/min; (**i**) 150 (**ii**) 300 (**iii**) 450 (**iv**) 600.

Figure 4 shows the effect of coolant flow rates (150–600 mL/min) and discharge rates (2 to 4 C) on the peak temperature and temperature difference across the battery at different coolant inlet temperatures (25 to 45 °C). At a low coolant flow rate (150 mL/min) and low inlet temperature (25 °C), the peak temperature and temperature difference (i.e., non-uniformity) across the battery surface are within the acceptable range ($T_{max}$ : $25 - 45$ °C; $\Delta T :< 5$ °C) at a 2 C discharge rate but the temperatures do not fall within the acceptable range at higher discharge rates (3–4 C). With the increase in the coolant flow rate from 150 mL/min to 300 mL/min, the flow in individual channels changes from laminar to turbulence resulting in enhanced heat transfer across the channel surface, which lowered the peak temperature as well as the temperature non-uniformity across the battery surface.

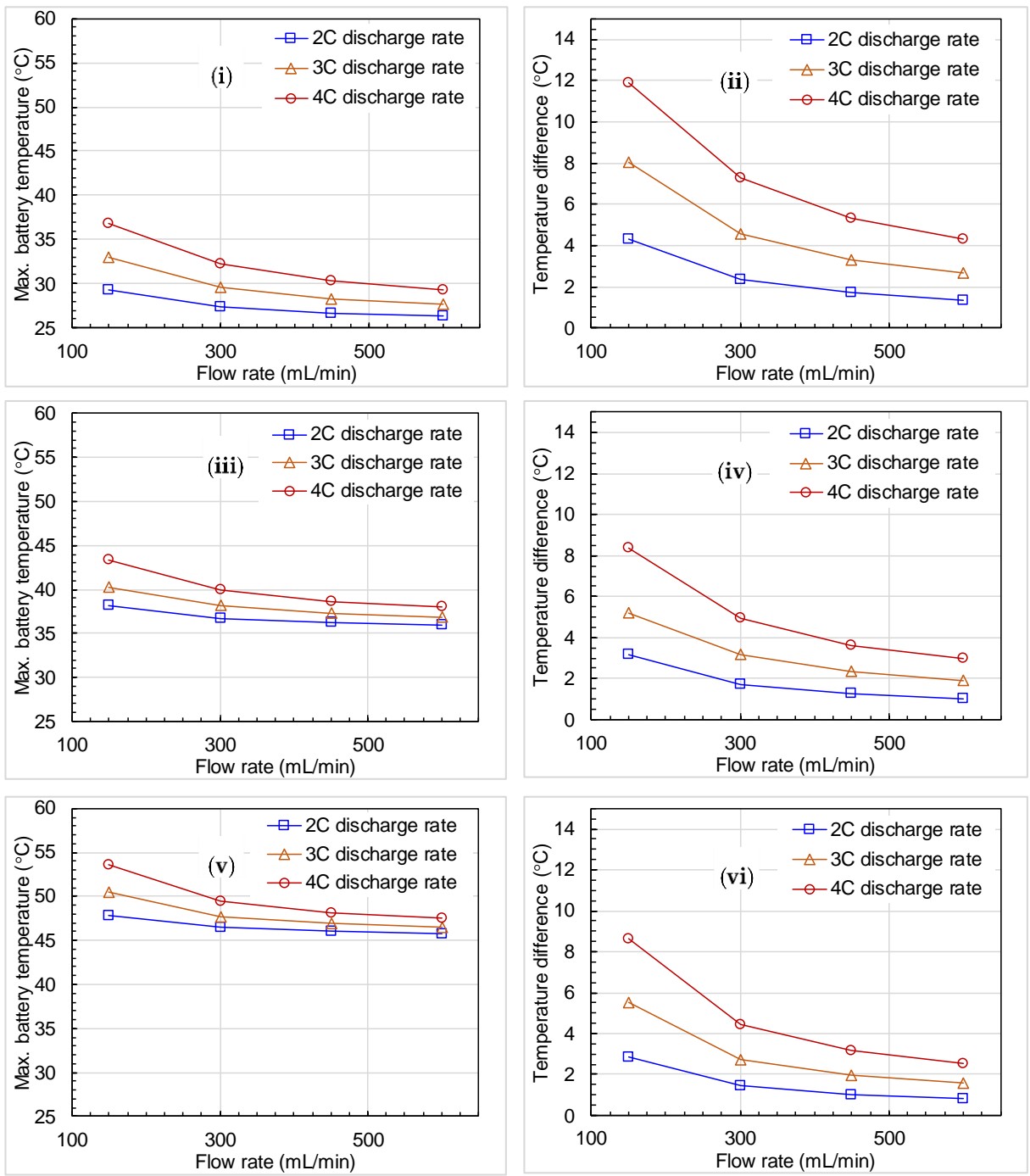

**Figure 4.** Predicted maximum battery surface temperature and temperature difference at different water flow rates and inlet temperatures; (**i,ii**) 25 °C (**iii,iv**) 35 °C (**v,vi**) 45 °C.

Further increase in the coolant flow rate from 300 to 450 mL/min lowered the battery temperatures which are within the acceptable range at all discharge rates and all coolant inlet temperatures. Overall, from Figure 4 it can be stated that the coolant flow rate has a high effect on the battery surface temperature non-uniformity at all discharge rates, while the coolant inlet temperature has a high effect on the battery surface peak temperature.

In order to study the effect of coolant conditions (inlet temperature and flow rate) on the battery surface temperature rise during discharge process, five different locations (*b, f, g, h, i*) have been particularly selected, as highlighted and shown in Figure 5. For a better understanding of the results, these locations are categorized as top and bottom (*b, i*) on

the mid vertical line, left and right (*f, h*) on the mid horizontal line and center (*g*) at the intersection of mid and horizontal lines on the battery surface.

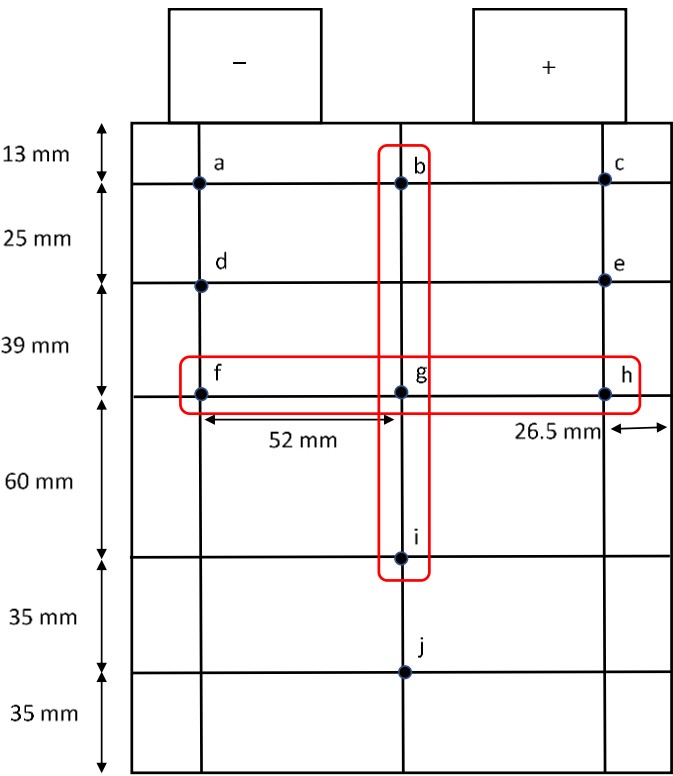

**Figure 5.** Selected locations (*b, f, g, h, i*) on the battery surface.

To test the fidelity of the coupled model (P2D), the predicted battery surface temperature rise at five different locations (as mentioned above) are compared with the measured data available with the present authors for a coolant flow rate of 150 mL/min (i.e., laminar convection). Figures 6–8 show a comparison of predicted and measured temperature rise during discharge at the top and bottom (*b, i*), left and right (*f, h*) and center (*g*) locations at two different coolant inlet temperatures (25–35 °C) and discharge rates (3–4 C).

At a low coolant inlet temperature (25 °C), the battery surface temperature rise during the discharge is steeper and is shown by the model and experiments, indicating that the heat generation is high at low temperatures. Further, with increasing the discharge rate, the temperature rise is steep at all five locations, which is obvious at a fixed coolant flow rate. It can be that from Figure 6 the model predicted temperatures at the top and bottom locations (*b, i*) and are either under predicted or over predicted as compared with the measured data. Similar trends are being observed at left and right locations (*f, h*), as shown in Figure 7. However, the predicted temperature rise at the center (*g*) of the battery surface are in close agreement with the measured data, as shown in Figure 8.

The little deviation between the predicted and measured data at top and bottom, and left and right locations can be attributed to the assumption of constant thermophysical property data used in Li-ion battery model available in COMSOL software, as well as the uniform volumetric heat generation throughout the battery during the discharge. Further, accuracy of the model input data used in the simulations may also partially contribute to the above disagreement.

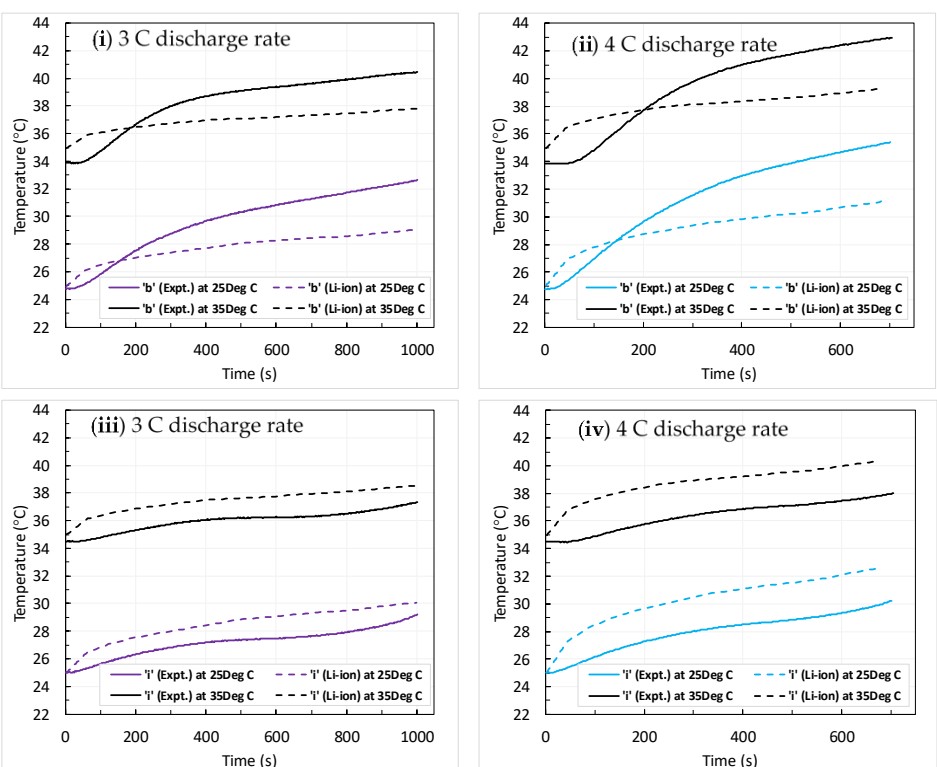

**Figure 6.** Predicted vs. measured battery surface temperature rise during discharge process at different coolant inlet temperatures; (**i**,**ii**) at top location; (**iii**,**iv**) at bottom location. Coolant flow rate fixed at 150 mL/min.

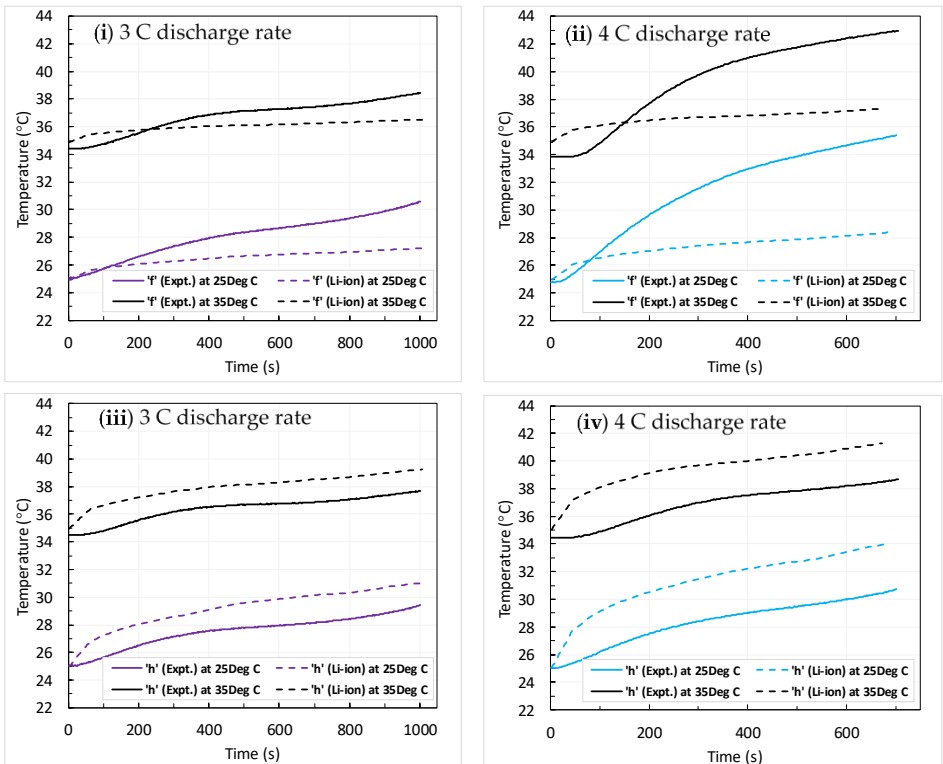

**Figure 7.** Predicted vs. measured battery surface temperature rise during discharge process at different coolant inlet temperatures; (**i**,**ii**) at left location (**iii**,**iv**) at right location. Coolant flow rate fixed at 150 mL/min.

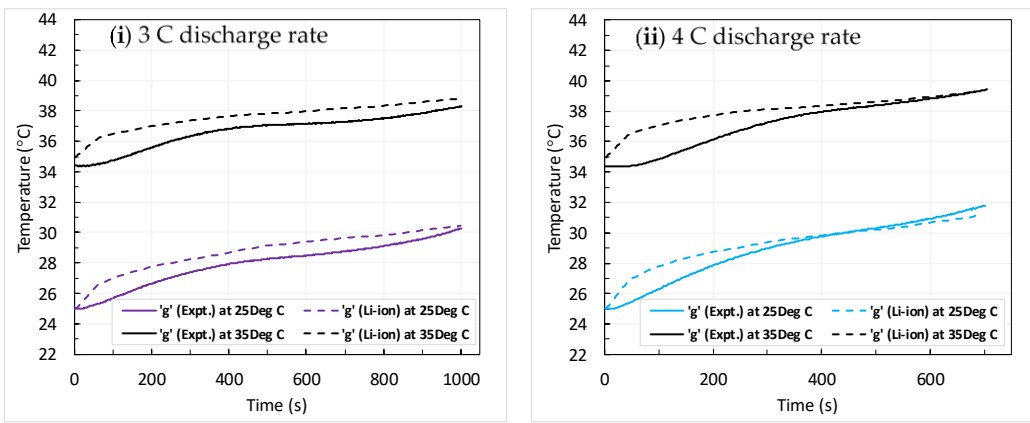

**Figure 8.** Predicted vs. measured battery surface temperature rise during discharge process at different coolant inlet temperatures; (**i,ii**) at center location. Coolant flow rate fixed at 150 mL/min.

Figure 9 shows the temperature rise at the top and bottom locations (*b*, *i*) of the battery surface at 3 C and 4 C discharge rates with different coolant inlet temperatures (25–45 °C) and flow rates (150–600 mL/min). It can be clearly observed that the battery surface temperatures increased faster with time at low coolant flow rate of 150 mL/min, which indicates that the heat generation by the battery is much higher than the heat dissipation by the coolant. Further, it is interesting to observe that at a low coolant inlet temperature (25 °C) and at higher coolant flow rates (i.e., turbulent convection), the rate of the rise in temperatures at the top and bottom locations are high, which indicates that the heat generation rate is high at low temperatures.

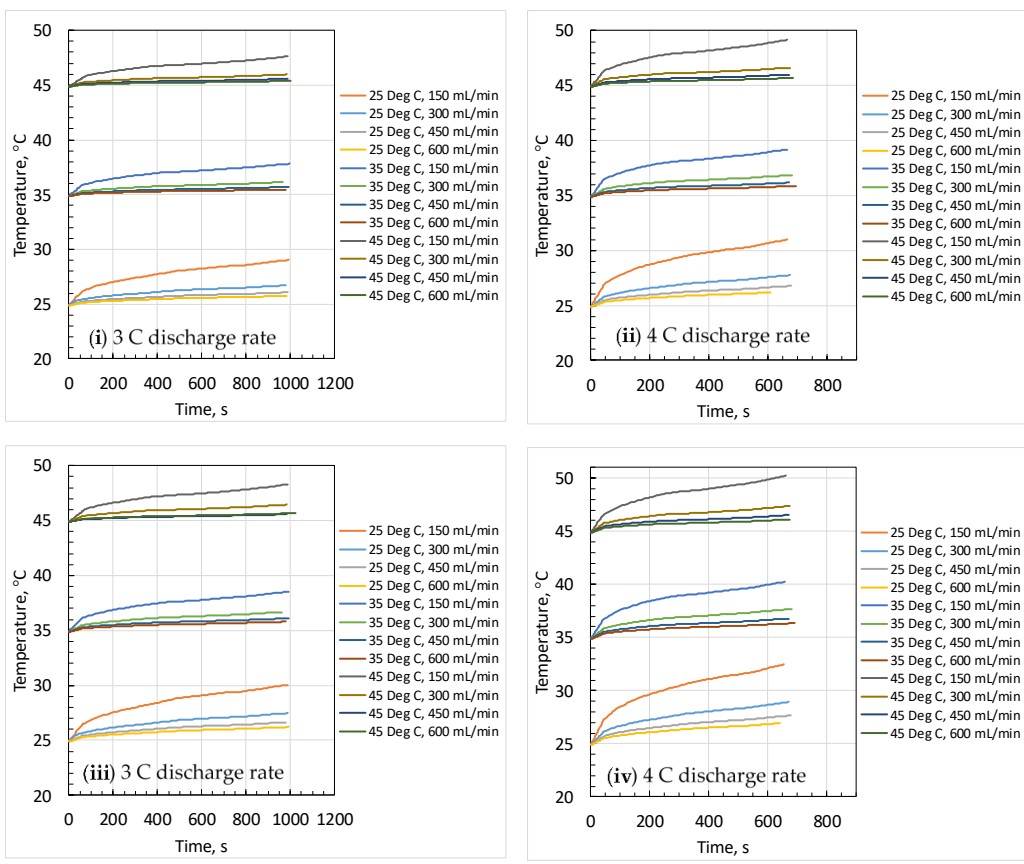

**Figure 9.** Predicted battery surface temperature rise during discharge process at different coolant flow rates and inlet temperatures; (**i,ii**) at top location; (**iii,iv**) at bottom location.

It can be noted that the low coolant inlet temperature keeps the battery at low temperatures by absorbing more heat. Additionally, the rise in temperatures at the left and right locations (*f*, *h*) shown in Figure 10 further strengthened the above argument because the left location is nearer to the coolant inlet where the battery surface was expected to be at low temperatures and still the rise is continuous and steep in contrast to that of medium and high coolant inlet temperatures and at all higher flow rates. Further, the rise in temperature during the discharge process at the center (*g*) of the battery shown in Figure 11 has a similar trend, which confirms that the above interpretation is correct.

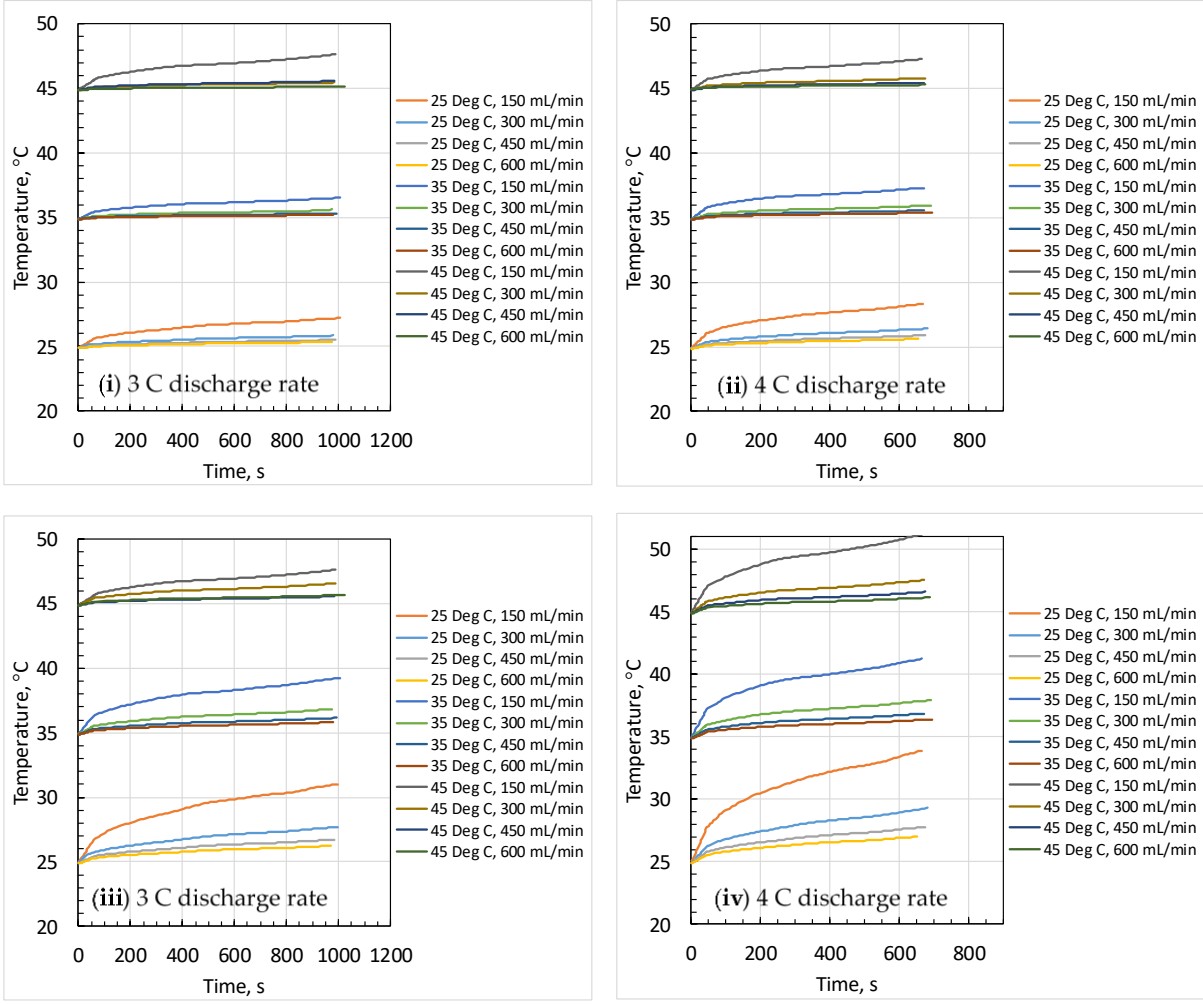

**Figure 10.** Predicted battery surface temperature rise during discharge process at different coolant flow rates and inlet temperatures; (**i**,**ii**) at left location; (**iii**,**iv**) at right location.

From Figures 9–11, it can also be noted that at high discharge rate (4 C), the battery surface temperature rise is steeper as compared to that at medium discharge rate (3 C), which is obvious. Further, the continuous and steep rise in battery surface temperatures also depend on the location because the coolant gets heated from inlet to the exit with less heat being absorbed, which is clearly observed with the right location (shown in Figure 10) at all coolant inlet temperatures and flow rates.

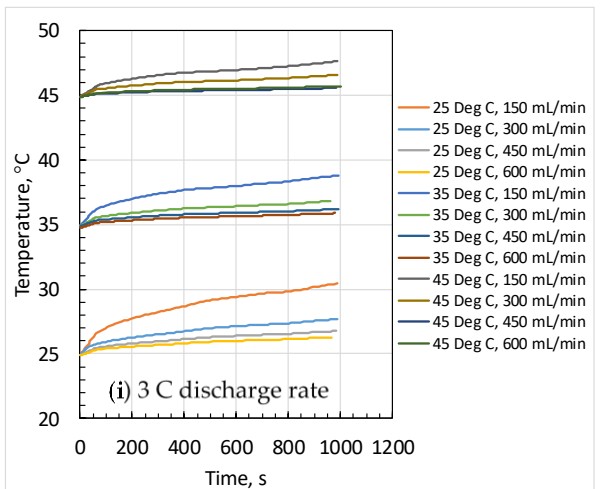 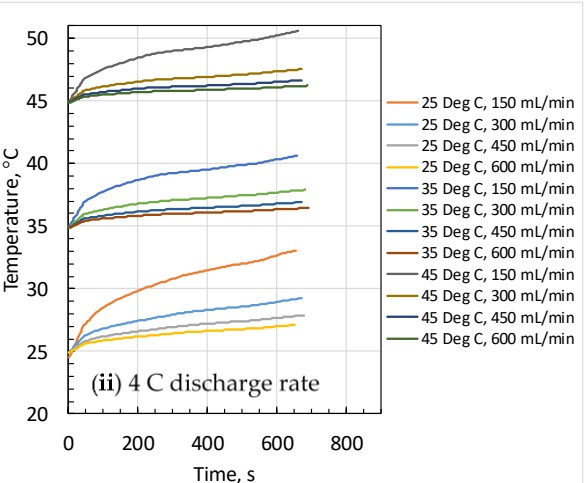

**Figure 11.** Predicted battery surface temperature rise during discharge process at different coolant flow rates and inlet temperatures; (**i,ii**) at center location.

## 4. Summary

A two-way coupled electrochemical-thermal model developed (in COMSOL software) by the present authors for 20 Ah LiFePO$_4$ prismatic battery placed between two mini-channel cold-plates has been used. The effects of water coolant inlet temperatures (25–45 °C) and flow rates (150–600 mL/min) on the effectiveness of a Battery Thermal Management System (BTMS) are studied. Transient simulations are performed at different discharge rates (2–4 C) and the predicted results are analyzed in terms of peak (maximum) temperature ($T_{max}$) and temperature difference ($\Delta T$) across the battery surface. Further, the model predicted a battery surface temperature rise at selected locations that has been validated with the experimental data for different coolant inlet temperatures and discharge rates. It can be noted that the flow in mini-channels is laminar at a flow rate of 150 mL/min, while the flow is turbulent at all other flow rates in the range of 300–600 mL/min. Therefore, it is expected that the enhanced heat dissipation with turbulent convection would lower the peak temperature as well as temperature non-uniformity. The following key observations are drawn from the present simulation-based study:

(i) The coolant flow rate has profound effect on the battery surface temperature uniformity, while the coolant inlet temperature has significant effect on the peak (maximum) temperature;

(ii) The coolant flow rate has effect on the battery surface peak temperature up to a certain limit, beyond which parasitic power requirements increase without a significant battery cooling;

(iii) A high coolant flow rate of 600 mL/min resulted in a drop in temperature difference across the battery surface by 6–10 °C from 2 C to 4 C discharge rates;

(iv) A low coolant inlet temperature of 25 °C and high coolant flow rate of 600 mL/min showed that the battery surface temperature rise is faster, indicating that the battery heat generation rate is high at a low coolant inlet temperature;

(v) At high coolant flow rates, the battery temperature difference is within the acceptable range ($\Delta T < 5$ °C), but the peak temperatures are not.

**Author Contributions:** Conceptualization, simulations and analysis, manuscript writing, and editing are performed by J.J., S.D.C. and C.A. Experiments and data formatting are performed by S.P. The overall works performed are supervised by J.J., M.F. and R.F. All authors have read and agreed to the published version of the manuscript.

**Funding:** This research received no external funding. The APC is supported by the co-author, M.F.

**Institutional Review Board Statement:** Not applicable.

**Informed Consent Statement:** Not applicable.

**Data Availability Statement:** Not applicable.

**Acknowledgments:** The authors from BITS Pilani, Hyderabad Campus would like to thank the institute management for providing the required computational resources.

**Conflicts of Interest:** The authors declare no conflict of interest.

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
