# Peer review of "Performance Study on the Effect of Coolant Inlet Conditions for a 20 Ah LiFePO4 Prismatic Battery with Commercial Mini Channel Cold Plates"

_2673-3293, doi:10.3390/electrochem3020018_

Round 1
Reviewer 1 Report
Reviewers' comments:
Manuscript number: electrochem-1691296
Title: Performance study on the effect of coolant inlet conditions for 20 Ah LiFePO4 prismatic battery with commercial minichannel cold-plates.
Comments:
The manuscript reported on Performance study on the effect of coolant inlet conditions for 20 Ah LiFePO4 prismatic battery with commercial minichannel cold-plates. The manuscript needs a detailed editing. It cannot be recommended for publication in the present form. I hope the following points would be helpful for the authors.
- In the Abstract: the authors need to improve with more specific short results and conclusions, i.e. academic novelty or technical advantages.
- The introduction section should be improved; more related papers must be discussed and superiority, novelty, critical improvement in this study must be clarified.
- Please provides the references for all equations and formula.
- Figure 7. Predicted vs. measured temperature variation during discharge at two coolant inlet temperatures and discharge rates; at centre (i, ii) of mid vertical and horizontal lines. Coolant flow rate fixed at 150 mL/min (laminar flow). – Not clear make clear.
- Figure 10 – not clear make clear.
- Several faults: are added or missing spaces between words: see manuscript file.
- Summary should be concise.
- References: there are recent references in 2021-2022 treating the same subject, you can use.
- Make all references in same format for volume number, page numbers and journal name, because it is difficult to searching and reading.
So that I recommended this manuscript to major revision and for future process.
Author Response
Dear Reviewer,
Greetings.
We, the authors, are grateful to you for the valuable comments and an opportunity given to us to revise the manuscript. Our responses to your comments are given below for your perusal. The required changes have been done and highlighted in the revised manuscript and to the best of our ability. The manuscript has improved significantly after the revision.
Thanks with regards,
Jeevan Jaidi

Reviewer 2 Report
In this manuscript, the authors have presented their numerical study of the heat generated during LFP-based lithium-ion battery cycling. The authors considered the influence of discharge rate and the fluid flow rate on the heat generated by the battery. The results are useful, and the paper is interesting to the journal readers. I would suggest publishing if the authors are able to address the following points.
Line 296-297: why there are no noticeable changes in the maximum temperature at a high flow rate?
The authors should explain why the maximum temperature is always at a far distance from the cathode.
What bases do the authors use to select the points of investigating the effect of coolant conditions on the surface temperature? I would think to appoint to the right of point J (where the maximum temperature always observed) would be useful
Author Response

(The authors gave the same response as above.)

Reviewer 3 Report
The paper presents an interesting study on the effect of inlet conditions in a BTMS. The results are clear and well presented. However, the description of the model is not clear enough and requires major revision. Please see the comments below.
Line 41 – You say battery peak temperature and temperature difference are the two critical parameters affecting performance. Can you say why? Or provide a reference to back this up.
Lines 51-120. Here you present some results from a literature study, but it feels a little disconnected and reads like a list of results. The paper would benefit from restructuring these paragraphs to give a better idea of the “story” of the paper – why is what you are doing important and how does it fit with existing work in the literature.
Line 148 – When you are discussing the BTMS I think it would help to include a sketch in this manuscript rather than referring the reader to previous work.
Lines 207-222 – The notation in the governing equations is not consistent or easy to follow. Again you give a reference to previous work, but it makes this manuscript hard to read. What are the subscripts 1,2? Also, many of the boundary conditions are missing or aren’t boundary conditions at all. For example for the equation for mass conservation of Li+ you say the boundary condition is D^{eff} = D \epsilon^{\gamma}, which is just the definition of the effective diffusivity. The presentation of the model equations needs major improvement.
Related to the above point – nowhere do you describe or give equations for how to calculate the heat generation terms from the battery model.
Line 242 – When you give the heat transfer model it is not explicitly stated which equations are being solved in the fluid domain (eq 7-17) and which in both the fluid and the battery (eq 18). It should be clearly stated where each of the equations holds. Again, the notation is inconsistent (you use a k-epsilon model, but these symbols have already been used to mean something entirely different in the battery model).
The following results are well presented and clear. They do a good job of describing the effect of inlet conditions as outlined in the abstract and introduction.
Line 355 – At this point, I became confused about the dimension of the battery model. Initially, I had assumed you were using a 3D (macroscopic) plus 1D (microscopic) battery model coupled to a 3D heat transfer problem, but at this point, it becomes clear the model is just the standard P2D model (i.e. 1 macroscopic dimension – the through-cell direction). This should be made more clear when the model is first presented. Given that you just have a P2D model can you comment on what effects your model doesn’t capture that could explain the top/bottom deviation? For example things like Ohmic drop in the current collectors, non-uniform discharge and the effect this has on heat generation, and how this heat generation feeds back into the electrochemistry.
There are lots of papers that model such effects, e.g. Kosch 2018 [Journal of The Electrochemical Society, 165 (10) A2374-A2388 (2018)], Kim 2007 [Journal of Power Sources 180 (2008) 909–916], Timms 2021 [https://doi.org/10.1137/20M1336898], Hosseinzadeh 2018 [doi:10.1016/j.jpowsour.2018.02.027]. Please note there is no expectation that any of these papers should be cited, they are simply suggestions of papers to take a look at.
Author Response

(The authors gave the same response as above.)

Round 2
Reviewer 1 Report
Now the paper is suitable for publication in this journal. It can be accepted.
Author Response
Kindly see the attachment for the changes made in the revised manuscript.

Reviewer 3 Report
Thank you for your careful revisions.
There are a couple of very minor issues remaining with the paper, but after these have been resolved the paper should be accepted for publication.
- You say "The boundary conditions used are same as in the COMSOL software module, and therefore not explicitly given in the manuscript". However, some boundary conditions are still given in section 2.2.1. Please either remove all boundary conditions or include all boundary conditions. And add a sentence pointing readers to where they can see the boundary conditions.
- You say "The instantaneous average heat generation is obtained from the battery (P2D) model for updating the battery temperature". Can you give equations or explicitly point to a reference where these terms are defined.
Well done on a very nice study!
Author Response

(The authors gave the same response as above.)
